# Nonlinear Dynamics Analysis of Disc Brake Frictional Vibration

**Hongyuan Zhang** *[ID], **Jiayu Qiao and Xin Zhang**

School of Automotive and Transportation, Shenyang Ligong University, Shenyang 110159, China
*   Correspondence: zhy_sylu@163.com

**Abstract:** The brake system is a key component to ensuring the safe driving and riding comfort of the vehicle, and the friction between the brake disc and the friction plate is the main source of vibration and noise. Therefore, in order to improve the stability of the braking system and reduce the generation of vibration, a six-degree-of-freedom nonlinear dynamics model was established, and using the Stribeck friction model and related parameters, the dynamic equation was solved by the Runge-Kutta method. The bifurcation diagram, Lyapunov diagram, time domain diagram, frequency spectrum diagram, and phase plane diagram of the brake pad and brake disc during friction braking were obtained, and the vibration characteristics of both under different braking pressure, braking speed, brake pad support stiffness, and brake disc support stiffness were analyzed. The results show that brake pressure is an important factor in triggering nonlinear vibration; increasing the braking speed will increase the amplitude of vibration, but will shorten the time to enter the stable motion state, and increasing the support stiffness brake pad and disc will reduce the amplitude of system vibration.

**Keywords:** disc brakes; vibration friction; dynamics analysis

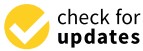



## 1. Introduction

The automotive braking process is a very complex tribological phenomenon. Disc brakes are widely used in many vehicles because of their superior performance. The braking force of the disc brake system comes from the friction between the brake disc and the brake pad. It is due to the nonlinearity of the contact stress at the contact interface and the nonlinearity of the relative motion speed of the friction pair. The frictional force excites the nonlinear vibration of the brake system, which not only affects the ride comfort of the vehicle, but also reduces the efficiency of the brake system and causes other failures. Therefore, the dynamics analysis based on reducing the frictional vibration characteristics of disc brakes is of great importance.

Many scholars have studied the nonlinear dynamics of the system caused by automotive braking friction. Sinou [1,2] studied transient and steady-state nonlinear self-excited vibrations and pointed out that linear stability analysis conditions are not valid in determining transient and steady-state oscillations, and that complex eigenvalue analysis does not predict such instabilities. Massi [3] conducted braking experiments on a typical friction block of a high-speed train. The friction and wear, interface temperature, vibration, and noise generated at the braking interface were studied, and the interrelationship between the vibration response and the interface contact behavior was analyzed. Ehret [4] found that the frictional force generated by the brake depends on the frictional characteristics between the brake disc and the brake pad and is influenced by the relative velocity, temperature, and normal pressure of the contact surfaces. Huemer [5] found that the impulsive excitation caused by the physical characteristics of the frictional contact is an important cause of the strong nonlinear vibrations in braking systems, axles, and chassis components. Elmaian [6] used the image-only model to describe the three main motion mechanisms of adhesion-slip, sliding, and modal coupling of the friction pair during the braking of a car, pointing out that the high-frequency whine originates from the modal coupling of the friction pair, the

twittering sound is caused by adhesion-slip, and both modal coupling and adhesion-slip cause the squeak. By applying a normal-loaded mass-damped-spring oscillator to a thin circular disc rotating at an equal speed, Zilin [7] found that when the rotating disc speed is low to a certain critical speed, the friction pad (mass block) and the brake disc (rotating disc) experience frequent separation-engagement alternations, when the system is destabilized and rich nonlinear dynamic characteristics appear. Others have developed dynamics models with different degrees of freedom for the problem of disc brake braking friction. Pan [8] considered a 6-degree-of-freedom brake system dynamics model using surface spring contact between the brake disc and brake pad, derived the corresponding kinetic differential equations, and constructed a Simulink model. The results show that the initial brake fluid pressure of the braking system has little effect on the magnitude of fluctuations in the braking torque, where as the proposed dynamics model has high accuracy in predicting braking vibrations. Yao [9] established a 6-degree-of-freedom nonlinear dynamics model for the disc brake vibration excitation problem and explored the effects of braking speed, braking pressure, and support stiffness on the vibration characteristics of the system. Ashley [10] analyzed a 4-degree-of-freedom torsional dynamics model of a disc brake system and found that the adhesion-slip discontinuity between the brake disc and the friction pad is the source of brake creep and gibberish. Wang [11] established a nonlinear torsional vibration model with four degrees of freedom and investigated its Lyapunov exponent characteristics and Hopf bifurcation properties. Shin [12] used a two-degree-of-freedom model in which the brake disc and brake pad were modeled as a single mode connected by a sliding friction interface. Using this model, the interaction between the brake disc and the brake pad was investigated. Shi [13] investigated the effect of different parameters of the system on the limit-loop characteristics by establishing a single-degree-of-freedom model combined with a hysteresis-loop friction model and plotting bifurcation and phase diagrams. The research methods and vibration characteristics analysis of other dynamical systems are also useful for this paper. Zhao [14] investigated the free-vibration behaviour of a functionally graded (FG) disk-shaft rotor system enhanced with graphene. A comprehensive parametric study of the effects of the weight fraction, distribution pattern, and aspect ratio of graphene nanosheets, as well as the effects of shaft length, elastic support stiffness, and rotational speed on the free vibration results was carried out in order to improve the vibration performance of the rotating FG-GPL disk-shaft. Zhao [15] investigated the coupled modelling approach and free vibration characteristics of a graphene nanosheet (GPL) enhanced blade-disk rotor system. The effects of rotational speed, GPL distribution pattern, GPL weight fraction, and inner and outer radius of the disc on the free vibration characteristics of the rotating blade shaft assembly are discussed in detail. Zhao [16] investigated the coupled model and vibration behaviour of rotating assembled cylindrical shell-plate structures. Modelling and vibration analyses were carried out by finite elements to investigate the effects of plate length-to-thickness ratio, plate width-to-thickness ratio, and plate position energy parameters on the travelling wave frequency of assembled cylindrical shell-plate structures. Zhou [17] took the ventilated disc brake as the research object, applied the finite element software ABAQUS, established the brake screech finite element model, carried out the complex eigenvalue analysis and the free mode analysis, and explored the relationship between the free mode and the friction coupling mode of the system components, the friction coefficient, and the influence of the elastic modulus of the brake disc brake pad, the key component of the brake system, on the braking stability.

The above study analyzed the relationship between nonlinear vibration characteristics and the friction contact interface, proposed that the instability of nonlinear self-excited vibration is unpredictable, found that the pulsed excitation caused by friction contact is an important cause of nonlinear vibration, established friction models with different degrees of freedom, and further explored the effect of different braking parameters on the vibration characteristics of the system. Moreover, the research methods of other dynamical systems are also helpful to the study of this paper. In this paper, a six-freedom nonlinear dynamics model for disc brakes is established, and the effect of braking speed on the

friction coefficient is considered when selecting the friction model. The bifurcation diagram, Lyapunov diagram, time domain diagram, frequency domain diagram, and phase plane diagram of brake pad and brake disc vibration were plotted by solving the model, and the effects of brake pressure, brake speed, brake pad support stiffness, and brake disc support stiffness on the vibration characteristics of the system were studied.

## 2. Dynamical Modeling

A typical disc brake consists of the caliper, brake pad, brake disc and piston, etc. Its structure and principle are shown in Figure 1. The brake calipers are fixed to the axle, and the friction pads are fixed to the support plate and are driven by pistons to apply braking force to the brake discs.

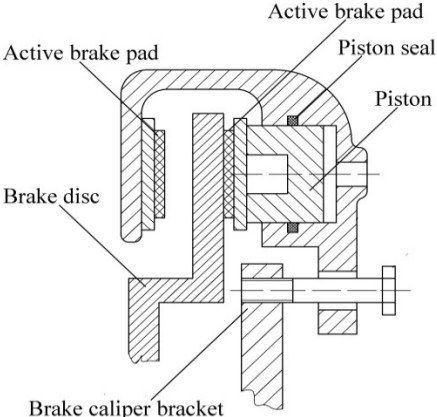

**Figure 1.** Schematic diagram of disc brake principle.

According to the installation and operation conditions of the disc brake shown in Figure 1, a six-degree-of-freedom dynamics model of the disc brake system was established, as shown in Figure 2. The x-axis is parallel to the brake disc plane and the y-axis is perpendicular to the brake disc plane. Where the fixed brake pad, brake disc, and movable brake pad masses are $m_1$, $m_2$ and $m_3$, respectively, and $m_1 = m_3$; the upper and lower brake pads have support stiffness $k_1$, $k_3$, damping $c_1$, $c_3$, and $k_1 = k_3$, $c_1 = c_3$, respectively; the normal support stiffness and damping between the fixed and movable brake pads and the brake disc are $k_{12}$, $c_{12}$, and $k_{23}$, $c_{23}$, respectively; the horizontal displacements are $x_1$, $x_2$, and $x_3$ respectively; the normal displacements are $y_1$, $y_2$, and $y_3$ respectively; $k_{2x}$, $k_{2y}$, and $c_{2x}$, $c_{2y}$ are the stiffness and damping between the brake disc and the support respectively; $P$ is the pressure applied to the brake pad by the wheel cylinder piston; $v$ is the initial velocity when the car starts braking. Based on the state of motion, the nonlinear vibration Equation (1) for six degrees of freedom is established.

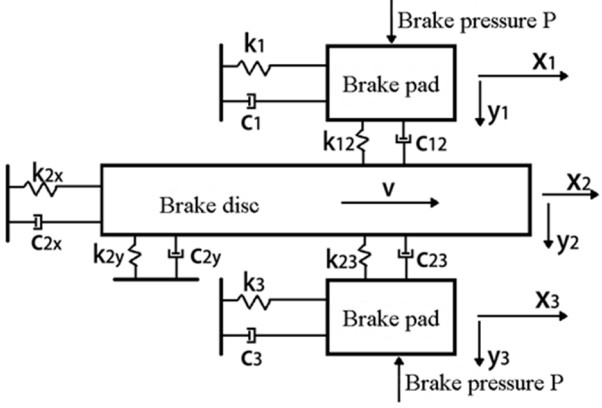

**Figure 2.** Dynamic model of disc brake system.

The system of equations are as follow:

$$\begin{cases} m_1\ddot{x}_1 + c_1\dot{x}_1 + k_1x_1 = \mu_1[k_{12}(y_1 - y_2) + c_{12}(\dot{y}_1 - \dot{y}_2)] \\ m_1\ddot{y}_1 = p - k_{12}(y_1 - y_2) + c_{12}(\dot{y}_1 - \dot{y}_2) \\ m_2\ddot{x}_2 + c_{2x}\dot{x}_2 + k_{2x}x_2 = -\mu_1[k_{12}(y_1 - y_2) + c_{12}(\dot{y}_1 - \dot{y}_2)] + \\ \mu_2[k_{23}(y_3 - y_2) + c_{12}(\dot{y}_3 - \dot{y}_2)] \\ m_2\ddot{y}_2 + c_{2y}\dot{y}_2 + k_{2y}y_2 = [k_{12}(y_1 - y_2) + c_{12}(\dot{y}_1 - \dot{y}_2)] + \\ [k_{23}(y_3 - y_2) + c_{12}(\dot{y}_3 - \dot{y}_2)] \\ m_3\ddot{x}_3 + c_3\dot{x}_3 + k_3x_3 = -\mu_2[k_{23}(y_3 - y_2) + c_{23}(\dot{y}_3 - \dot{y}_2)] \\ m_3\ddot{y}_3 = -p + k_{23}(y_3 - y_2) + c_{23}(\dot{y}_2 - \dot{y}_3) \end{cases} \tag{1}$$

where $\mu_1$ is the friction coefficient of the upper fixed friction pad and the brake disc, and $\mu_2$ is the friction system of the movable friction pad and the brake disc.

The simplified equations are as follow:

$$\begin{cases} m_1\ddot{x}_1 + c_1\dot{x}_1 - \mu_1c_{12}\dot{y}_1 - \mu_1c_{12}\dot{y}_2 + k_1x_1 - \mu_1k_{12}y_1 + \mu_1k_{12}y_2 = 0 \\ m_1\ddot{y}_1 + c_{12}\dot{y}_1 - c_{12}\dot{y}_2 + k_{12}y_1 - k_{12}y_2 = p \\ m_2\ddot{x}_2 + c_{2x}\dot{x}_2 + \mu_1c_{12}\dot{y}_1 + (\mu_2c_{12} - \mu_1c_{12})\dot{y}_2 - \mu_2c_{12}\dot{y}_3 + k_{2x}x_2 + \\ \mu_1k_{12}y_1 + (\mu_2k_{23} - \mu_1k_{12})y_2 - \mu_2k_{23}y_3 = 0 \\ m_2\ddot{y}_2 - c_{12}\dot{y}_1 + (c_{2y} + 2c_{12})\dot{y}_2 - c_{12}\dot{y}_3 - k_{12}y_1 + (k_{12} + k_{23} + k_{2y})y_2 - k_{23}y_3 = 0 \\ m_3\ddot{x}_3 + c_3\dot{x}_3 - \mu_2c_{23}\dot{y}_2 + \mu_2c_{23}\dot{y}_3 + k_3x_3 - \mu_2k_{23}y_3 + \mu_2k_{23}y_2 = 0 \\ m_3\ddot{y}_3 - c_{23}\dot{y}_2 + c_{23}\dot{y}_3 - k_{23}y_2 + k_{23}y_3 = -p \end{cases} \tag{2}$$

The dynamic equation of the whole braking system are as follows:

$$[M]\{\ddot{X}\} + [C]\{\dot{X}\} + [K]\{X\} = \{F\} \tag{3}$$

$$[M]\{\ddot{Y}\} + [C]\{\dot{Y}\} + [K]\{Y\} = \{F\} \tag{4}$$

where the displacement vector is $\{X\} = \{x_1, x_2, x_3\}^T$, $\{Y\} = \{y_1, y_2, y_3\}^T$. The force vector is $\{F\} = \{0, p, 0, 0, 0, -p\}^T$.

The quality matrix is as follows:

$$[M] = \begin{bmatrix} m_1 & & \\ & m_2 & \\ & & m_3 \end{bmatrix}_{3\times3} \tag{5}$$

The damping matrix is as follows:

$$[C] = \begin{bmatrix} c_1 & & & & & \\ & c_{12} & & & & \\ & & c_{2x} & & & \\ & & & c_{2y} & & \\ & & & & c_{23} & \\ & & & & & c_3 \end{bmatrix}_{6\times6} \tag{6}$$

The stiffness matrix is as follows:

$$[K] = \begin{bmatrix} k_1 & & & & & \\ & k_{12} & & & & \\ & & k_{2x} & & & \\ & & & k_{2y} & & \\ & & & & k_{23} & \\ & & & & & k_3 \end{bmatrix}_{6\times 6} \tag{7}$$

## 3. Dynamic Analysis of Disc Brakes

The random vibration generated during the friction braking of a vehicle is directly related to the selection of each parameter of the braking system, so it is necessary to study the influence of the braking parameters of the random vibration of the system. This section uses the controlled variable method to study the effects of brake pressure, brake speed, and support stiffness on system stability in turn. It provides a reference for improving the comfort of vehicle operation and the stability of the braking system.

Numerous studies have shown that the coefficient of friction between the brake disc and brake pad is non-linear with the sliding speed of both. After reviewing the information, the coefficient of friction with the growth of vehicle speed is in a decreasing trend, and there is a certain pattern to follow. It fits in the following Stribeck friction model [14].

$$\mu = -\mu_s \text{sgn}(v_r) + \alpha v_r - \beta v_r^3 \tag{8}$$

where, $\alpha = 3(\mu_s - \mu_m)/(2v_m)$, $\beta = (\mu_s - \mu_m)/(2v_m^3)$. $v_r$ is the relative sliding speed of the brake disc and the brake pad. $v_m$ is the velocity corresponding to the minimum kinetic friction coefficient $\mu_m$. $\mu_s$ is the coefficient of static friction. Refer to the literature to take [14], $v_m = 0.35$, $v_r = v_m = 90$ km/h, $\mu_s = 0.57$.

The model is written in the form of a system of equations as follows:

$$\begin{cases} \mu_1 = -\mu_s \text{sgn}(v_{r1}) + \alpha v_{r1} - \beta v_{r1}^3 \\ \mu_2 = -\mu_s \text{sgn}(v_{r2}) + \alpha v_{r2} - \beta v_{r2}^3 \\ v_{r1} = \dot{x}_1 - v - \dot{x}_2 \\ v_{r2} = \dot{x}_3 - v - \dot{x}_2 \end{cases} \tag{9}$$

### 3.1. Effect of Brake Pressure on the Nonlinear Characteristics of the System

In this paper, by numerically solving the system of Equation (1) with the fourth-order Runge-Kutta method, we first obtain the time-domain vibration response of the system, then plot the bifurcation, Lyapunov, and phase diagrams of the system response using the multiset solution, and finally obtain the frequency-domain diagram of the system response using the fast Fourier transform method. By reviewing the relevant literature, the stiffness and damping coefficients of the disc brake system were obtained, as shown in Table 1. Because the upper and lower brake pads have the same kinematic characteristics and are the axial linear systems, there will be no stable limit ring. The following content only analyzes the tangential motion of the upper brake pads and brake discs and calls the upper brake pads brake pads. The motion bifurcation diagram of brake pads and brake discs was drawn. As shown in Figure 3, when the braking pressure is 1000 N and the braking speed is between 0 km/h and 50 km/h, the brake pads and discs produce random vibration. As shown in Figure 4, when the braking pressure is 6000 N and the braking speed is between 0 km/h and 95 km/h, the brake pads and discs exhibit strong chaotic characteristics. As the brake pressure increases, the amplitude of the brake pad and disc vibration increases significantly, and the corresponding braking speed increases significantly when the system is in chaotic motion. The results show that excessive brake pressure is one of the important factors that induce nonlinear vibration in the system.

**Table 1.** Disc brake dynamics parameters.

| Parameter | Value |
|---|---|
| $k_1, k_2, k_{2x}, k_{2y}/(\text{N·m}^{-1})$ | $2.5 \times 10^7$ |
| $k_{23}, k_{12}/(\text{N·m}^{-1})$ | $3.9 \times 10^7$ |
| $c_1, c_3, c_{2x}, c_{2y}/(\text{N·s·m}^{-1})$ | 295 |
| $c_{23}, c_{12}/(\text{N·m}^{-1})$ | 495 |
| $m_1, m_2/\text{kg}$ | 0.135 |
| $m_3/\text{kg}$ | 8.33 |

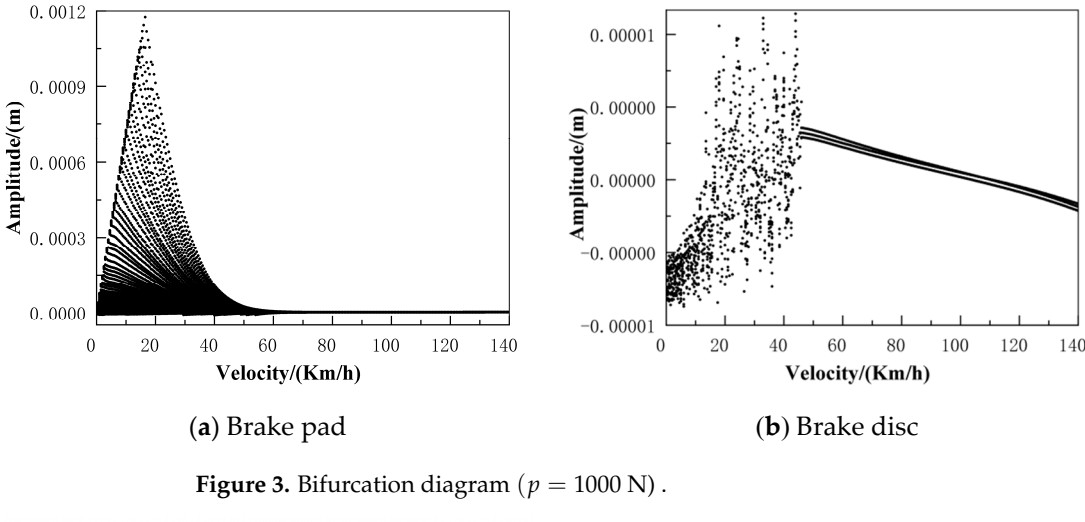

(**a**) Brake pad      (**b**) Brake disc

**Figure 3.** Bifurcation diagram ($p = 1000$ N).

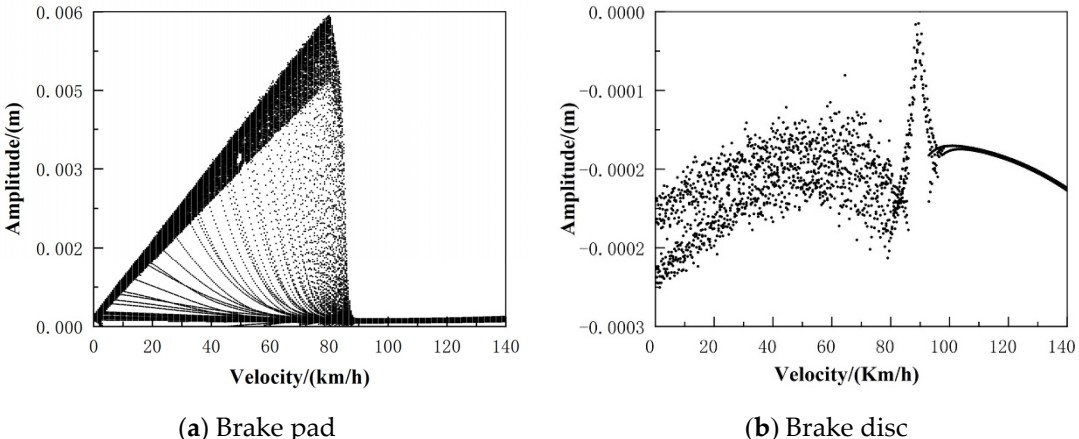

(**a**) Brake pad      (**b**) Brake disc

**Figure 4.** Bifurcation diagram ($p = 6000$ N).

The Lyapunov exponent is a numerical value used to describe the divergence of neighbouring trajectories in space. When the Lyapunov exponent is less than 0, it indicates that the system is converging in that direction; when the Lyapunov exponent is greater than 0, it indicates that the system is diverging in that direction [18]. With the Lyapunov index, the state of motion of the system can be determined directly. The Lyapunov diagram of the whole system under different braking pressures is drawn, as shown in Figure 5. Under different braking pressures, the Liapunov exponent of the system is always greater than 0, indicating that the system is always in a chaotic motion state. When the braking pressure is 1000 N and the braking speed is between 0 km/h and 50 km/h, the Lyapunov exponent varies randomly and then tends to be stable; when the braking pressure is 6000 N and the braking speed is between 0 km/h and 80 km/h, the Lyapunov exponent varies randomly and then tends to be stable, and the random vibration characteristics of the system are enhanced. Therefore, the higher the braking pressure, the stronger the vibration characteristics of the system.

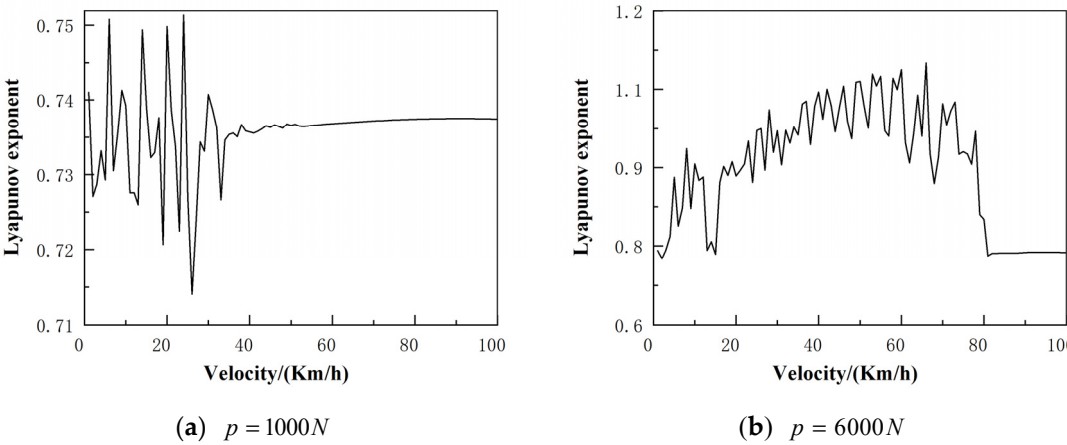

(**a**) $p = 1000N$  (**b**) $p = 6000N$

**Figure 5.** Lyapunov diagram of the brake system under different brake pressures.

The braking speed was selected as 60 km/h, and the time domain, spectrum, and phase plane diagrams of the brake pads and discs under different braking pressures were further compared and analyzed. As shown in Figures 6 and 7, as the braking pressure increases, the time for the brake pad to enter a state of equal amplitude vibration decreases and the amplitude of the vibration becomes larger; the vibration amplitude of the brake disc also increases, and it is always in an unstable motion, with irregular time domain curve and irregular vibration.

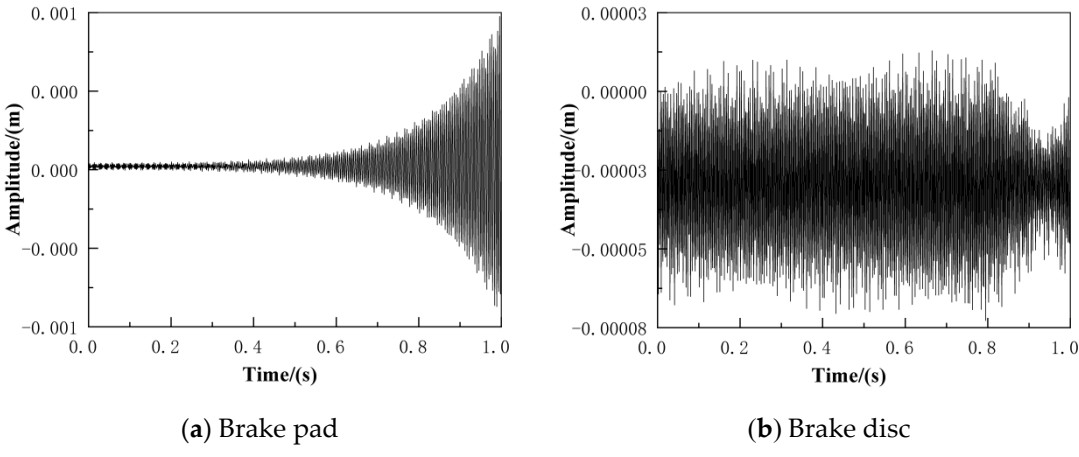

(**a**) Brake pad  (**b**) Brake disc

**Figure 6.** Time domain diagram ($p = 1000$ N) .

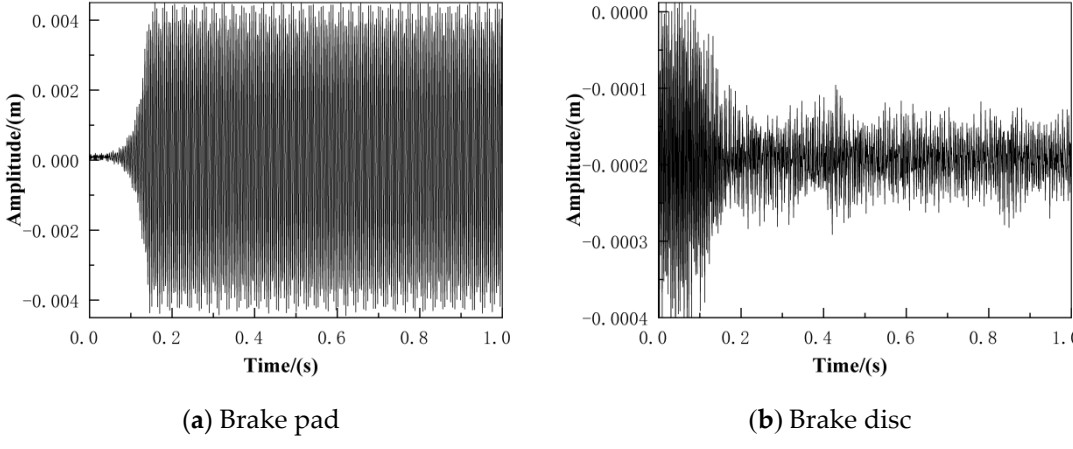

(**a**) Brake pad  (**b**) Brake disc

**Figure 7.** Time domain diagram ($p = 6000$ N) .

As shown in Figures 8 and 9, in the range of 0–500 Hz, when the brake pressure is 1000 N, the main frequency of brake pad vibration is 210 Hz, with the increase in brake pressure, the main frequency of vibration is gradually prominent, and the amplitude of vibration increases; the main frequency of brake disc vibration is 300 Hz, and as the braking pressure increases, other vibration frequencies different from the main frequency gradually stand out, the amplitude of vibration increases, and the chaotic characteristics of the system are enhanced.

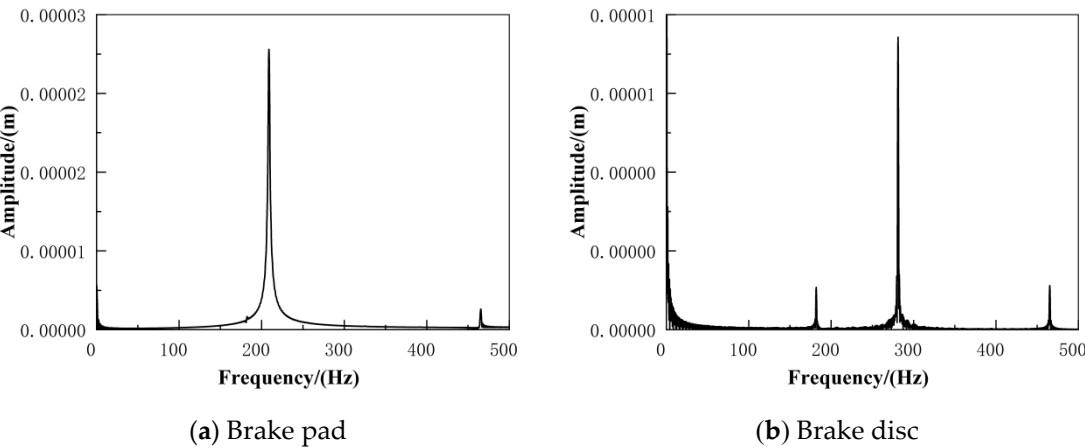

(**a**) Brake pad                                    (**b**) Brake disc

**Figure 8.** Frequency spectrum diagram ($p = 1000$ N) .

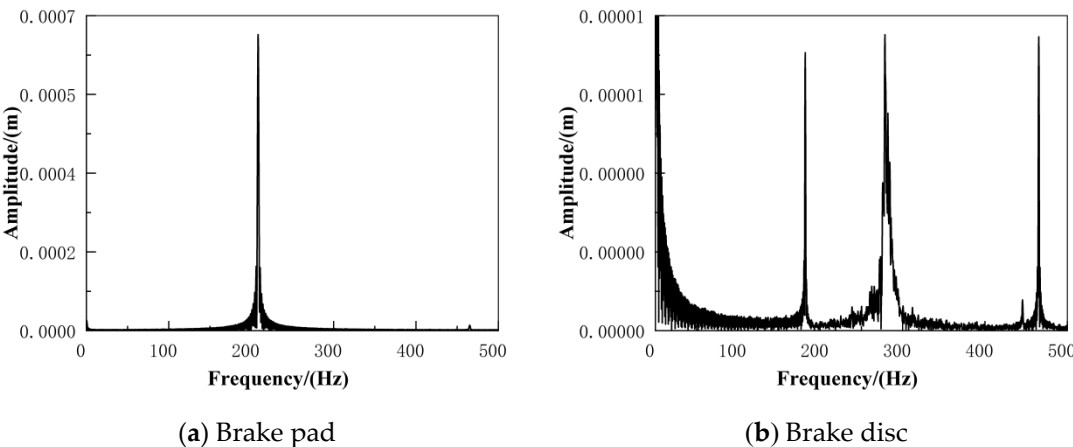

(**a**) Brake pad                                    (**b**) Brake disc

**Figure 9.** Frequency spectrum diagram ($p = 6000$ N) .

The phase trajectory line represents the process of changing the system state, as shown in Figures 10 and 11; when the brake pressure is small, both the brake pad and the brake disc are in a relatively stable state of motion. With the increase in braking pressure, the brake pad enters the viscous-slip motion state from the pure sliding motion state, and the amplitude of the phase diagram increases; the phase trajectory line of the brake disc is more irregular, the motion state is more complicated, and the random vibration characteristics of the system are enhanced. The results show that brake pressure is one of the important factors affecting the random vibration of the system.

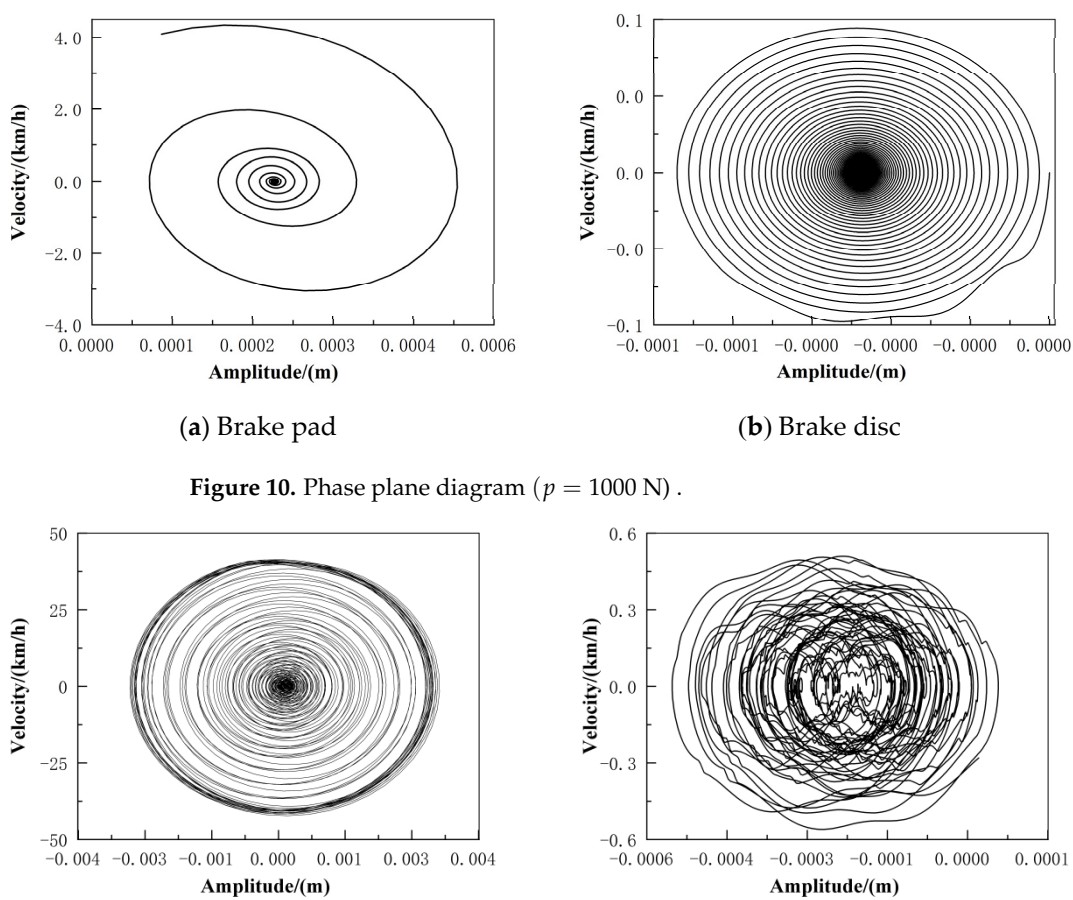

(**a**) Brake pad                    (**b**) Brake disc

**Figure 10.** Phase plane diagram $(p = 1000 \text{ N})$.

(**a**) Brake pad                    (**b**)Brake disc

**Figure 11.** Phase plane diagram $(p = 6000 \text{ N})$.

*3.2. Effect of Initial Braking Speed on the Nonlinear Characteristics of the System*

The bifurcation of the brake pad and disc movements at different braking speeds were plotted, as shown in Figures 12 and 13. With the increase in braking speed, the amplitude of brake pad vibration increases, and the chaotic region in the figure increases; the vibration characteristics of the brake disc are similar to those of the brake pad, and the system vibration characteristics are enhanced. The Lyapunov diagram of the system at different braking speeds was plotted, as shown in Figure 14. At different braking speeds, the maximum Lyapunov exponent of the system is greater than 0 and the system is in a chaotic state of motion.

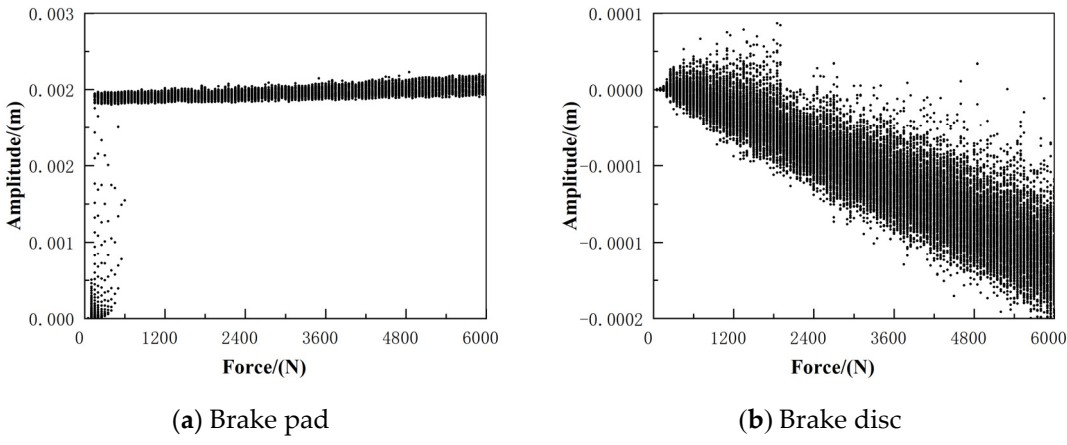

(**a**) Brake pad                    (**b**) Brake disc

**Figure 12.** Bifurcation diagram (v = 10 km/h).

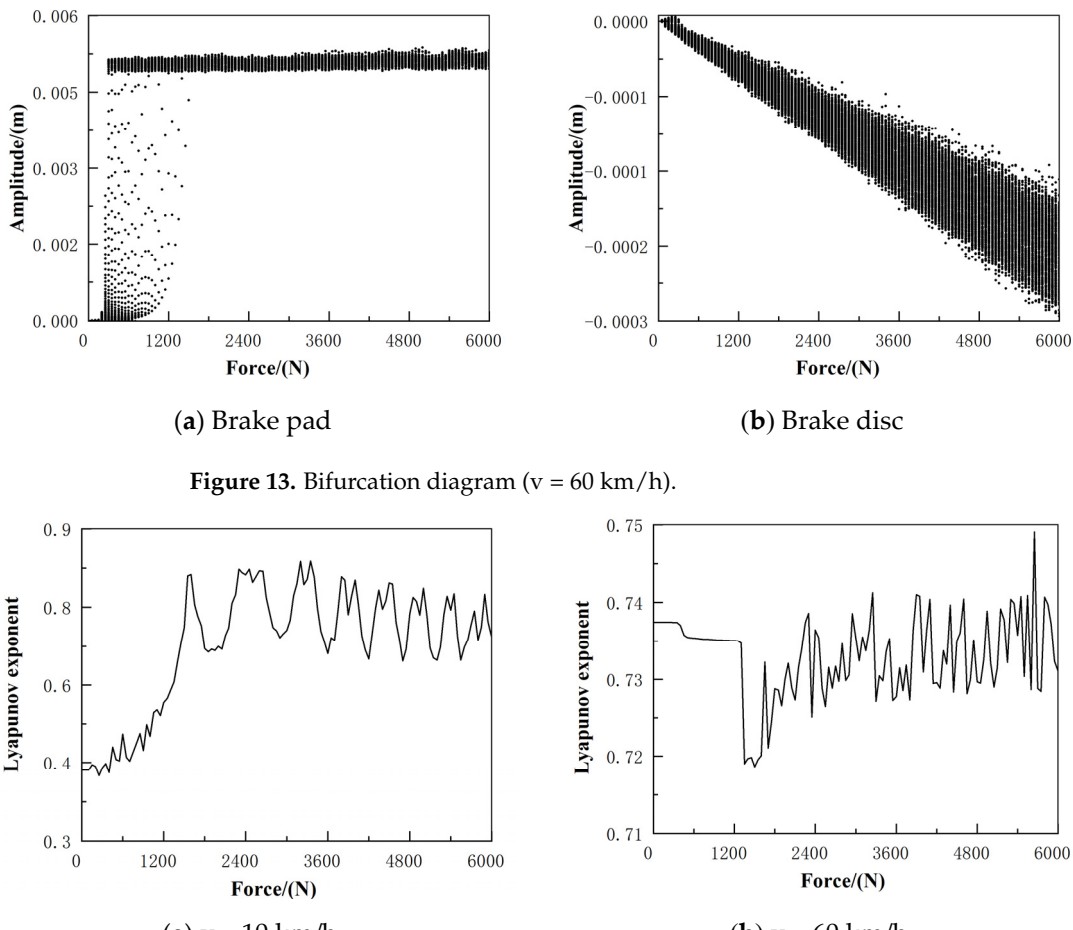

**Figure 13.** Bifurcation diagram (v = 60 km/h).

**Figure 14.** Lyapunov diagram of the braking system at different braking speeds.

The time domain diagrams of the brake pad and disc movements at different braking speeds were plotted for a selected braking pressure of 4000 N, as shown in Figures 15 and 16. When the braking speed is 10 km/h, the brake pad and disc appear to vibrate randomly, and with the increase in time, both eventually reach a steady state of motion; when the braking speed is 60 km/h, the brake pad and disc gradually change from an unstable state at first to a stable state. As the braking speed increases, the time for the system to reach the stable state becomes shorter and shorter. However, the amplitude of both brake pad and brake disc showed a tendency to increase with speed, and the vibration characteristics of the system were enhanced and the vibration was more violent.

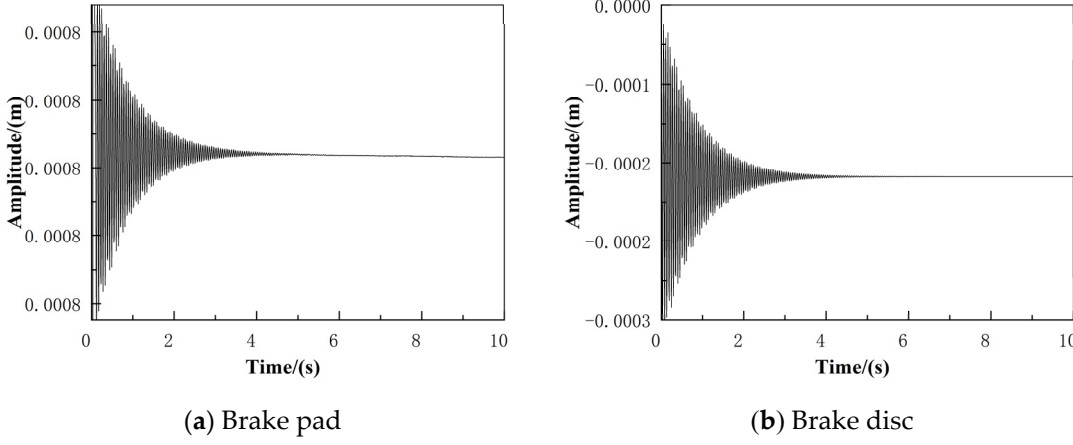

**Figure 15.** Time domain diagram (v = 10 km/h).

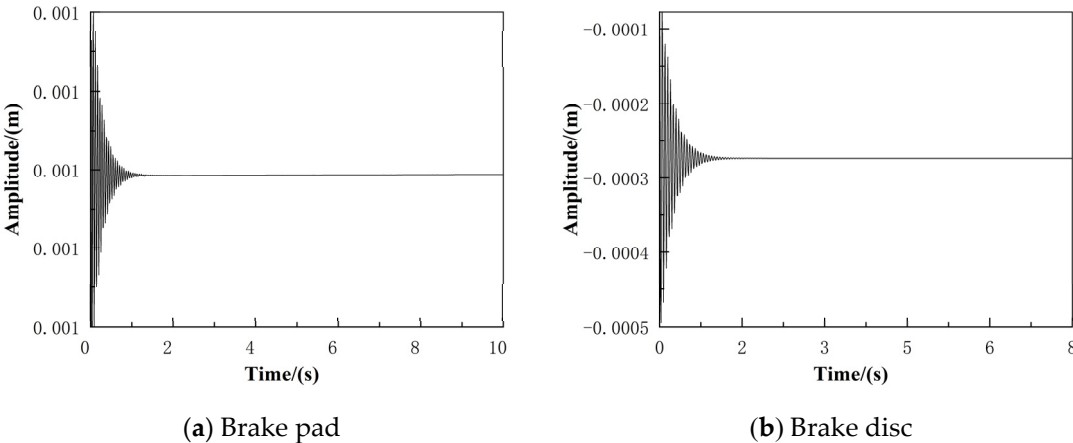

(**a**) Brake pad        (**b**) Brake disc

**Figure 16.** Time domain diagram (v = 60 km/h).

The frequency spectrum diagrams of the brake pad and disc movements at different braking speeds were plotted for a selected braking pressure of 4000 N, as shown in Figures 17 and 18. When the initial braking speed is 10 km/h, the brake pad and brake disc produces different vibrations in the range of 0–500 Hz; when the initial braking speed is 60 km/h, the vibration frequencies of both gradually concentrate to the main frequency, and the main vibration frequency gradually stands out and the amplitude gradually increases, and the random vibration characteristics of the system increase. This is because as the initial braking speed gradually increases, the stick-slip vibration cycle of the system shortens, and the slope of the brake pad displacement growth during the sticking phase becomes greater. When the speed increases to a certain level, the viscous slip motion gradually evolves into a pure slip motion, and the vibrations that are different from the main frequency disappear.

The braking pressure was chosen to be 4000 N, and the phase plane diagrams were plotted in turn for initial braking speeds of 10 km/h and 60 km/h. As shown in Figures 19 and 20, when the braking speed is 10 km/h, the phase trajectory line is confused, and the brake pad and brake disc display obvious stick-slip phenomena; when the braking speed is 60 km/h, the brake pad and disc are already in the stable movement stage. With the increase in braking speed, the phase trajectory gradually becomes orderly and neat, and the system is in a relatively stable state of motion; at the same time, the number of turns of the phase trajectory line gradually decreases, indicating that the time for the system to reach stable motion decreases. The results show that increasing the braking speed increases the amplitude of the system vibration, but shortens the time for the system to enter the steady state.

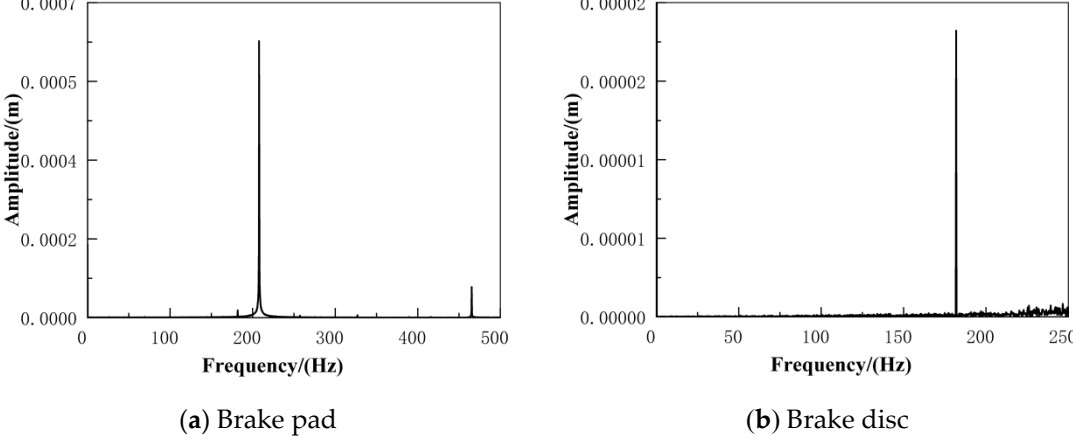

(**a**) Brake pad        (**b**) Brake disc

**Figure 17.** Frequency spectrum diagram (v = 10 km/h).

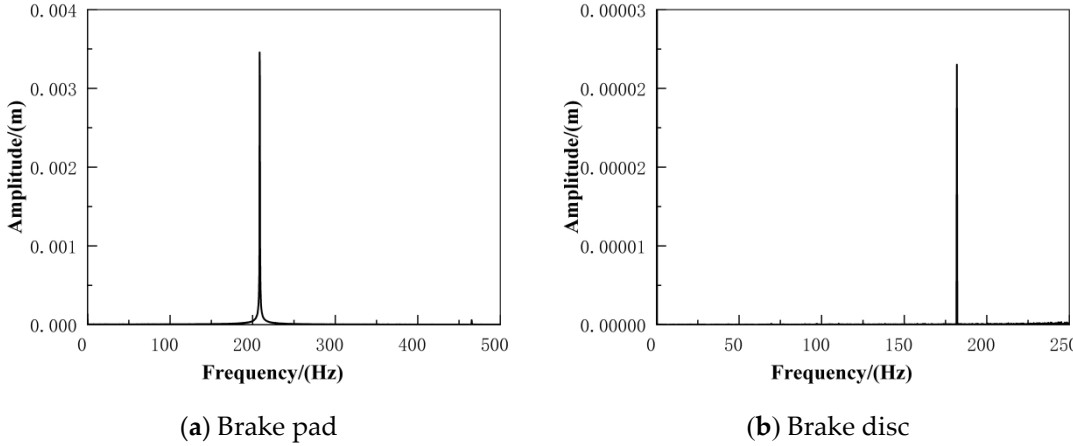

(**a**) Brake pad        (**b**) Brake disc

**Figure 18.** Frequency spectrum diagram (v = 60 km/h).

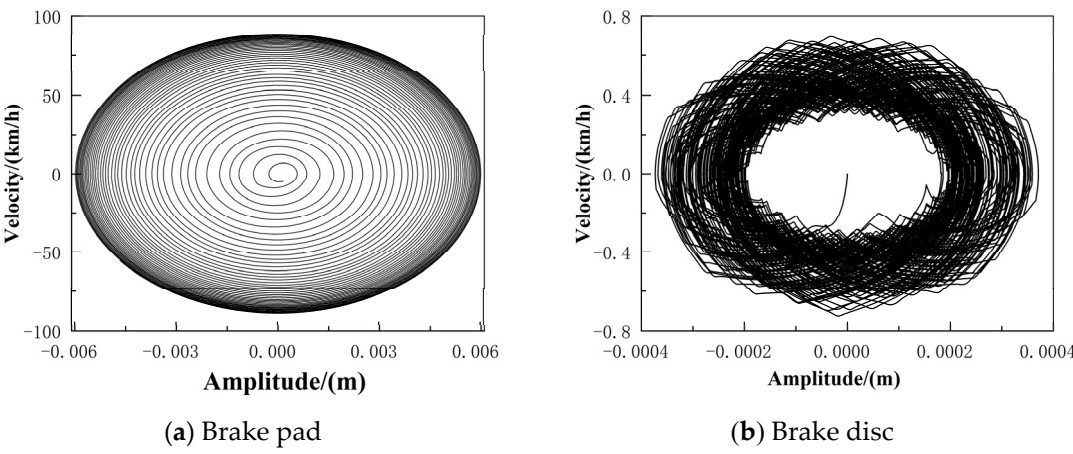

(**a**) Brake pad        (**b**) Brake disc

**Figure 19.** Phase plane diagram (v = 10 km/h).

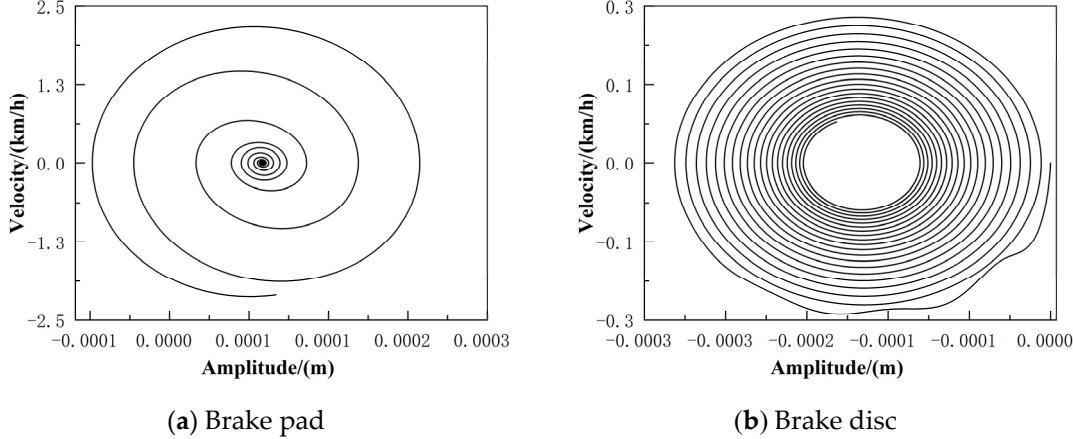

(**a**) Brake pad        (**b**) Brake disc

**Figure 20.** Phase plane diagram (v = 60 km/h).

### 3.3. Effect of Brake Pad Support Stiffness on the Non-Linear Characteristics of the System

The stiffness of the structure is particularly important when the car is in motion, as it determines the safety, reliability, and comfort of the vehicle, among other indicators. This section uses the controlled variable method to analyze the effect of the support stiffness of the brake pad on its system vibration. The braking pressure is selected as 4000 N, the initial braking speed is 40 km/h, and the system stiffness and damping coefficients are shown in Table 2, to study the effect of different brake pad stiffness on system stability.

**Table 2.** System dynamics characteristics parameters (change the brake pad support stiffness).

| Parameter | Value |
|---|---|
| $k_{2x}, k_{2y}/(\text{N·m}^{-1})$ | $2.5 \times 10^7$ |
| $k_{23}, k_{12}/(\text{N·m}^{-1})$ | $3.9 \times 10^7$ |
| $c_1, c_3, c_{2x}, c_{2y}/(\text{N·s·m}^{-1})$ | 295 |
| $c_{23}, c_{12}/(\text{N·m}^{-1})$ | 495 |

Figure 21 shows the bifurcation of the system vibration motion as the brake pad support stiffness $k_1$ and $k_3$ change from $1 \times 10^7$ N/m to $6 \times 10^7$ N/m. As the brake pad support stiffness increases, the chaotic characteristics of the brake pad diminish, effectively suppressing the random vibration of the brake pad, but with little effect on the brake disc, so this section only discusses the effect of the brake pad support stiffness on the vibration characteristics of the brake pad.

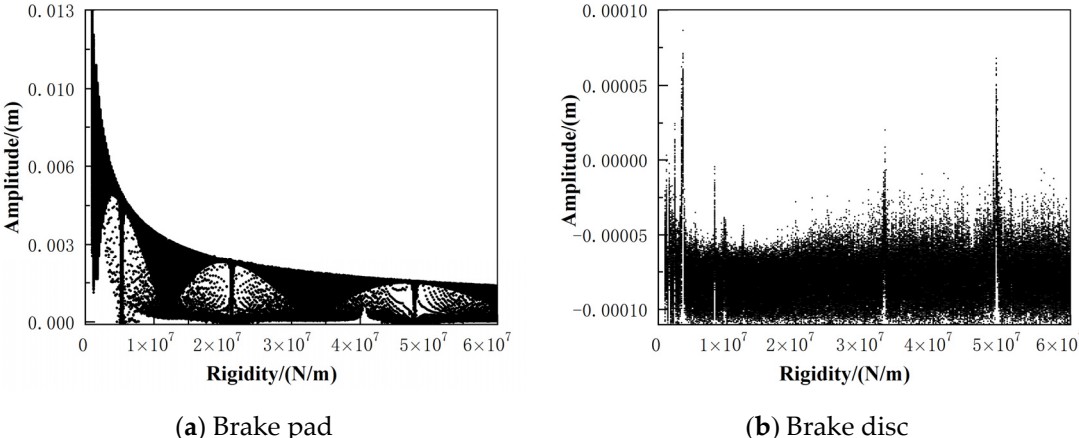

(**a**) Brake pad
(**b**) Brake disc

**Figure 21.** Bifurcation diagram of system motion when brake pad stiffness changes.

The time domain and frequency spectrum of brake pad at different support stiffness were plotted. As shown in Figure 22, as the support stiffness increases, the amplitude of brake pad vibration decreases from 0.004 m to 0.002 m, indicating that increasing the support stiffness of the brake pad can suppress the random vibration characteristics of the brake pad. As can be seen from Figure 23, the main frequency of the brake pad does not change significantly as the support stiffness increases, and the amplitude decreases with the increase in support stiffness from 0.005 m to 0.0002 m.

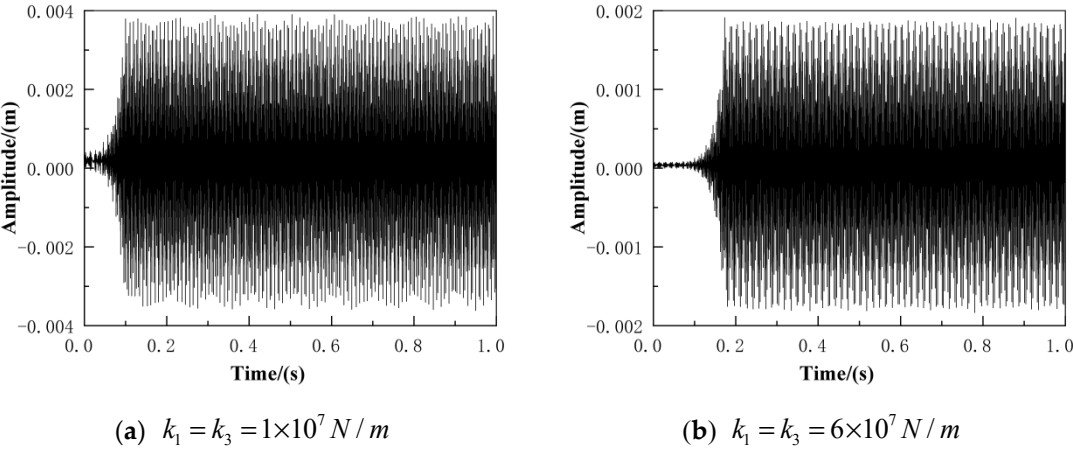

(**a**) $k_1 = k_3 = 1 \times 10^7 \, N/m$
(**b**) $k_1 = k_3 = 6 \times 10^7 \, N/m$

**Figure 22.** Brake pad time domain diagram.

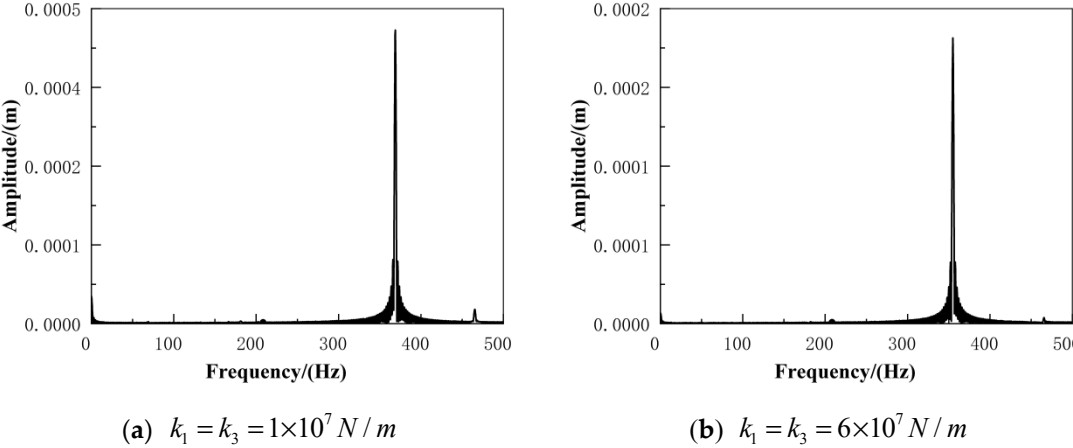

**(a)** $k_1 = k_3 = 1 \times 10^7 \, N/m$　　　　　　　**(b)** $k_1 = k_3 = 6 \times 10^7 \, N/m$

**Figure 23.** Brake pad frequency spectrum diagram.

The phase plane of the brake pad as the pad support stiffness changes were plotted, as shown in Figure 24. As the brake pad support stiffness increases, the range of phase plane trajectory lines gradually decreases and the random vibration characteristics of the system diminish. Consequently, the brake pad support stiffness can be selected in a relatively high range.

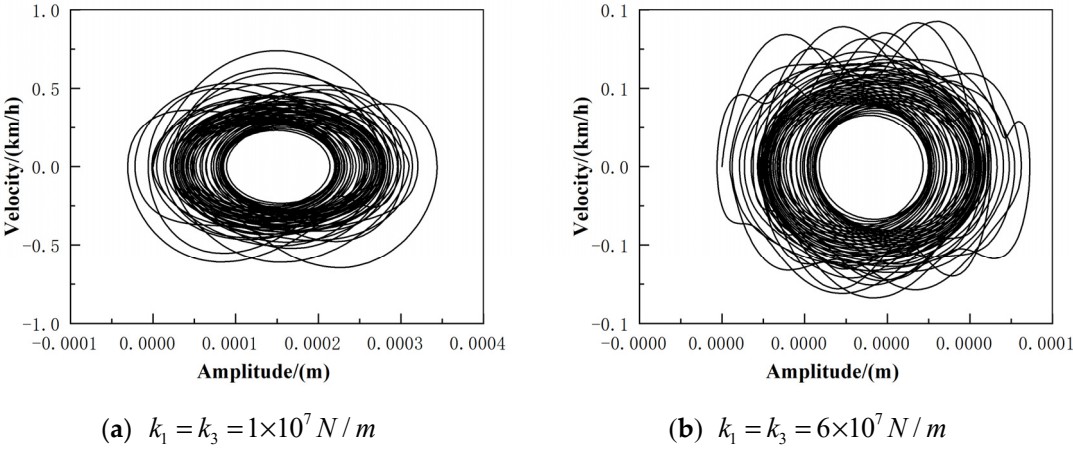

**(a)** $k_1 = k_3 = 1 \times 10^7 \, N/m$　　　　　　　**(b)** $k_1 = k_3 = 6 \times 10^7 \, N/m$

**Figure 24.** Brake pad phase plane diagram.

### 3.4. Effect of Brake Disc Support Stiffness on the Non-Linearcharacteristics of the System

The effect of brake disc support stiffness on the non-linear vibration characteristics of the system was investigated. The braking pressure is selected as 4000 N, the braking speed is 40 km/h, and the stiffness and damping coefficient of the system are shown in Table 3.

**Table 3.** System dynamics characteristics parameters(change the brake disc support stiffness).

| Parameter | Value |
| --- | --- |
| $k_1, k_3 / (\text{N·m}^{-1})$ | $2.5 \times 10^7$ |
| $k_{23}, k_{12} / (\text{N·m}^{-1})$ | $3.9 \times 10^7$ |
| $c_1, c_3, c_{2x}, c_{2y} / (\text{N·s·m}^{-1})$ | 295 |
| $c_{23}, c_{12} / (\text{N·m}^{-1})$ | 495 |

The bifurcation diagram of brake disc and brake pad vibration movement under different brake pad support stiffness was drawn. As shown in Figure 25, increasing the stiffness of the brake disc reduces the chaotic characteristics of the brake disc, but has little

effect on the brake pad, so this section only discusses the effect of the brake pad support stiffness on the vibration characteristics of the brake pad.

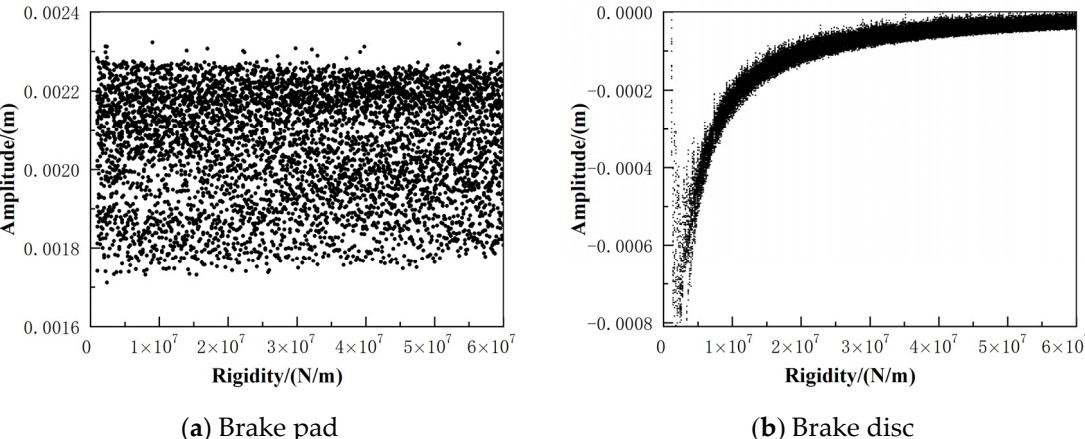

(**a**) Brake pad    (**b**) Brake disc

**Figure 25.** Bifurcation diagram of system motion when brake disc stiffness changes.

Figure 26 shows the time versus displacement curve as the brake disc support stiffness varies. As the support stiffness increases, the amplitude of the system vibration decreases and the time to enter steady motion is shortened.

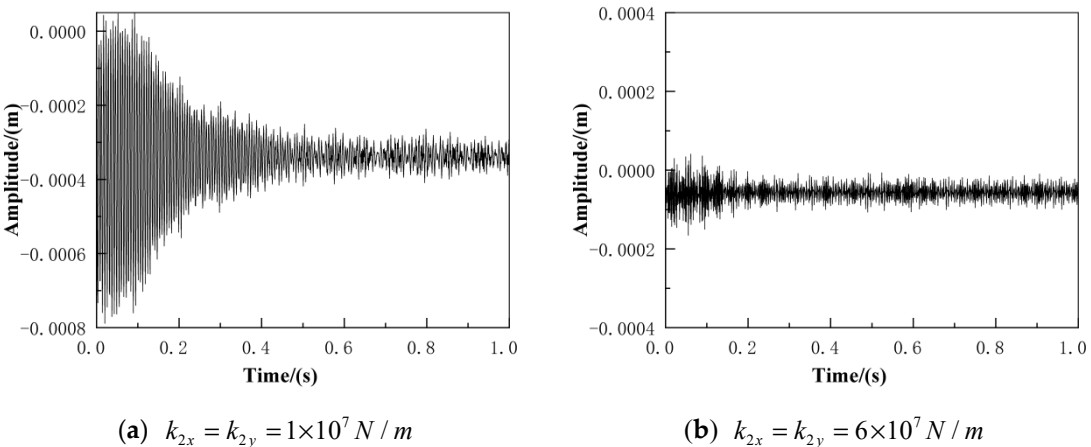

(**a**) $k_{2x} = k_{2y} = 1 \times 10^7\, N\,/\,m$    (**b**) $k_{2x} = k_{2y} = 6 \times 10^7\, N\,/\,m$

**Figure 26.** Brake disc time domain diagram.

Figure 27 shows the frequency versus amplitude of the brake pad and disc vibration for different disc stiffness. When the support stiffness is $1 \times 10^7$ N/m, the main frequency of friction plate vibration reaches 170 Hz and increases to 240 Hz when the stiffness reaches $6 \times 10^7$ N/m. More high-frequency vibrations are also generated. The supporting stiffness of the brake disc should therefore be chosen to be in the medium range.

Figure 28 shows the phase plane of the brake disc as the support stiffness of the disc changes. When the support stiffness is small, the quasi-static point of the system is unstable; as the support stiffness increases, the quasi-equilibrium point of the disc begins to move inwardly until it encounters the inwardly stable purified sliding limit ring and then gradually merges into the stable limit ring, at which point the system is the combined motion of the stable equilibrium point and the stable limit ring.

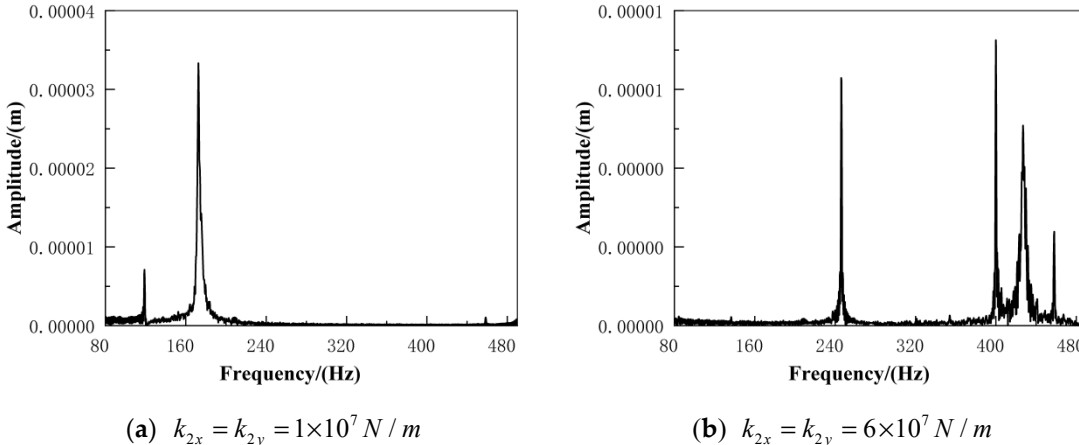

(**a**) $k_{2x} = k_{2y} = 1 \times 10^7 \, N / m$      (**b**) $k_{2x} = k_{2y} = 6 \times 10^7 \, N / m$

**Figure 27.** Brake disc frequency spectrum diagram.

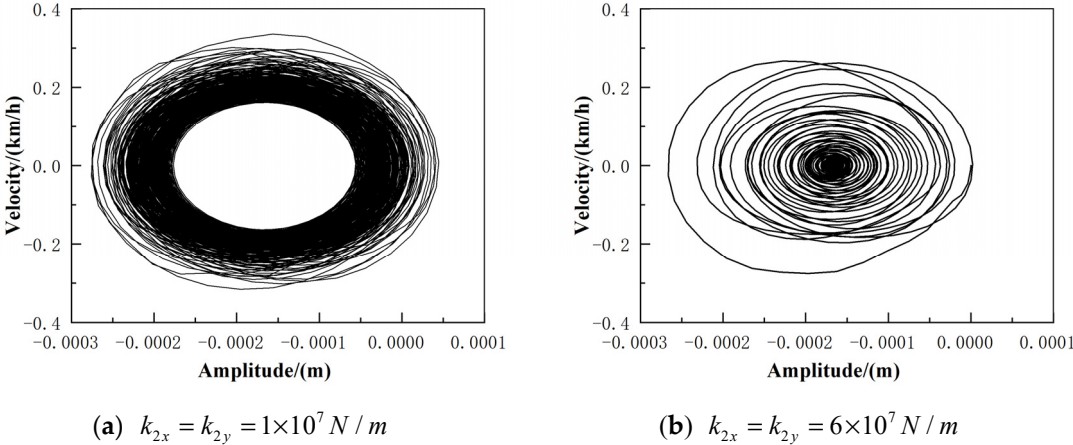

(**a**) $k_{2x} = k_{2y} = 1 \times 10^7 \, N / m$      (**b**) $k_{2x} = k_{2y} = 6 \times 10^7 \, N / m$

**Figure 28.** Brake disc phase plane diagram.

### 4. Conclusions

In this paper, a six-degree-of-freedom non-linear dynamics model of a disc brake is established. By solving the dynamics model, the non-linear vibration characteristics of the disc brake are obtained under different braking conditions and different dynamics characteristic parameters.

1. When the braking pressure is low, the system is in a relatively stable state of motion. As the braking pressure increases, the brake pads undergo a process of moving from a stable to a stick-slip state; the brake discs go directly from a stable state to an unstable vibration state, and the vibration becomes more complex and irregular. As the brake pressure increases, the vibration amplitude of the brake pads and discs also increases, and the intensity of the vibration becomes greater. Braking pressure is one of the most important factors causing random vibration in the system.

2. When the braking speed is low, the system is in a stick-slip motion. As the braking speed increases, the system gradually enters a state of pure sliding motion. This is caused by the negative slope relationship between friction and velocity, the negative slope being equivalent to a negative damping. At low braking speeds, the negative damping of the system is greater than the positive damping, and the system gathers energy, thus generating self-excited vibrations; as the braking speed increases, the negative damping of the system is less than the positive damping and the system consumes energy, eventually reaching a steady state. The results show that increasing the braking speed increases the amplitude of the system vibration but shortens the time for the system to enter steady motion.

3. As the brake pad support stiffness increases, the vibration of the brake pad can be suppressed. Consequently, the brake pad support stiffness should be taken to a higher range.

4. When the brake disc stiffness is small, the random vibration characteristics of the system are strong. As the brake disc support stiffness increases, the amplitude of the brake disc vibration gradually decreases, and the random vibration characteristics of the system are weakened, but this will lead to other high-frequency vibration of the brake disc, so the brake disc support stiffness should be selected within a medium range; the quasi-equilibrium point of the brake disc gradually moves towards the inner measurement, and eventually, the system carries out the integrated movement of the stable equilibrium point and the stable limit ring.

**Author Contributions:** H.Z. was mainly responsible for the organization and structure definition, and provided the initial review of the contribution. J.Q. was to the highest extent responsible for the draft version preparation, figures, and literature research. X.Z. introduced ideas, suggested the organization of the work, and was also responsible for proofreading. All authors have read and agreed to the published version of the manuscript.

**Funding:** This research was funded by Shenyang Youth Science and technology innovation talent project (RC200006), Scientific research fund project of Liaoning Provincial Department of Education (LG202008), Liaoning Province Basic Research Projects of Higher Education Institutions (Grant No. LG202107), the construction plan of scientific research and innovation team of Shenyang Ligong University (SYLU202101), Comprehensive reform project of graduate education of Shenyang Ligong University (2021DSTD004, 2021PYPT006).

**Institutional Review Board Statement:** Not applicable.

**Informed Consent Statement:** Not applicable.

**Data Availability Statement:** Not applicable.

**Conflicts of Interest:** The authors declare no conflict of interest.

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
