# Peer review of "Nonlinear Dynamics Analysis of Disc Brake Frictional Vibration"

_applsci, doi:10.3390/app122312104_

Round 1
Reviewer 1 Report
The manuscript is of quite limited novelty and significance. This paper will be more appropriate for a conference paper.
I am advising against publication of your work at present for the following reasons:
- the novelties of the manuscript with the state of the art are not highlighted clearly. Many works on both the stability analysis and nonlinear dynamics of friction-induced vibration have been proposed (see the well-known studies of Massi et al., Sinou et al., Hoffmann et al., Ouyang et al., etc).
- the numerical simulations are based on classical approaches without original contribution.
- the modeling of the brake system is also very simple and the numerical results, as well as the proposed analysis are not new. Consequently, discussion are too poor to warrant publication in Applied Sciences (inferior quality)
- comparison between experiments should be done.
Because of the above, the original contribution is considered insufficient for publication. The scientific content is of inferior quality and without great interest for the scientific community working in the field of friction-induced vibration and noise.
I realise that this decision will be very disappointing for the authors, however I hope that you find these comments constructive for further refinement of your work.
Reviewer 2 Report
The author have established and studied six-degree-of-freedom non-linear dynamics model of a disc brake. However, the significance of this in the real application is weakly addressed which need to be strengthened. Along with that in the majority of the sections grammatical errors were observed which need to be rectified.
Reviewer 3 Report
In this study, the authors investigate the brake disc model of the 6-th order. They show that when simulating the dynamics of the system by the 4-th order Runge-Kutta method, a nonlinear motion is observed. To refine the parameters at which it occurs, the authors plot bifurcation diagrams, phase portraits, and waveforms.
The overall research is interesting and actual. Nevertheless, some questions and remarks arise while reading the manuscript.
1. The main research flaw, in my opinion, is that the conclusions are only of a qualitative nature. Meanwhile, authors could plot the optimal speeds on the braking force, or get other numerical dependencies useful in engineering practice.
2. In addition, it would be interesting to see experimental confirmation that the dynamics of a real brake match the model. As is known from many studies on nonlinear dynamics, for example, DOI: 10.1007/s11071-022-07854-0 or DOI: 10.3390/app11010081, there may be significant differences between numerical model and its real-world counterpart.
3. References section contains works only from 2020-2022, not including earlier studies. I suppose this requires improving. Moreover, the review part of the manuscript in page 2 lists the researches without any structuring. Are there any groups of approaches to studying vibration during braking? What the information about this vibration is used for: to implement control systems, to increase the reliability of mechanical break parts, to increase the comfort of passengers in transport? Authors should clarify.
4. The authors claim that the motion under study is chaotic. Use Wolf Lyapunov exponent estimation or the simpler method proposed in DOI: 10.1142/S0218127416502266 to find the largest Lyapunov exponent (LLE) and confirm that the motion is indeed chaotic. Also, plotting the LLE against speed and force is a fine idea. Waveforms (Fig. 5 and similar) with a higher time resolution also would be useful.
5. The manuscript is not written very carefully and contains errors and typos.
Two first sentences are by mistake repeated 2 times.
The sentence in Lines 28-33 is too long and requires splitting into 2 or 3 parts.
In line 146, 'isas' replace to 'is as'
and some other typos. Authors should check the overall text.
I think that the paper may be accepted after the major revision.
Round 2
Reviewer 1 Report
Dear author. as previously explained in my first review, the manuscript is of quite limited novelty and significance. This paper will be more appropriate for a conference paper.
I am advising against publication of your work at present for the following reasons:
- Many works on both the stability analysis and nonlinear dynamics of friction-induced vibration have been proposed. See papers on FIVN in the international journals JSV, MSSP, Tribology International, etc...
- There is no experimental validation. You must be aware that there are a large number of papers on this subject, many of which are not robust and thus would fare poorly when experimental data are used. Comparison with experiments is needed to validate simulation.
- the numerical simulations are based on classical approaches without original contribution.
- the modeling of the brake system is too simple and the numerical results, as well as the proposed analysis are not new. Again no original contribution is proposed.
This contribution is considered insufficient for publication.